# Impact of mixed-species forest plantations on soil mycobiota community structure and diversity in the Congolese coastal plains

**Lydie-Stella Koutika[1,2]\***, **Arthur Prudêncio de Araujo Pereira[3]**, **Alessia Fiore[4]**, **Silvia Tabacchioni[4]**, **Manuela Costanzo[4]**, **Luciana Di Gregorio[4]**, **Annamaria Bevivino[4]\***

1 Centre de Recherche sur la Durabilité et la Productivité des Plantations Industrielles (CRDPI), Pointe-Noire, Republic of the Congo, 2 Soil Care and Environmental Studies (SCES), Pointe-Noire, Republic of the Congo, 3 Soil Science Department, Federal University of Ceará, Fortaleza, Ceará, Brazil, 4 Department for Sustainability, Italian National Agency for New Technologies, Energy and Sustainable Economic Development, ENEA Casaccia Research Centre, Rome, Italy

\* ls_koutika@yahoo.com, lydie.stella.koutika@gmail.com (LSK); annamaria.bevivino@enea.it (AB)

**Data Availability Statement:** Raw sequence data reported in this study have been deposited in the National Center for Biotechnology Information

## Abstract

Mixed tree plantations containing nitrogen ($N_2$)-fixing species have the potential to enhance C sequestration, soil biodiversity and forest productivity. Here, we investigated the impact of *Acacia mangium* and *Eucalyptus urophilla x E. grandis* mixed plantations in the Congolese coastal plains on soil mycobiota community structure and diversity by ITS metabarcoding sequencing and bioinformatic analysis. Higher Faith's phylogenetic diversity and Evenness' was found in Eucalyptus monoculture relative to stands containing Acacia. Differences in beta diversity were found among Eucalyptus and Acacia monoculture, and mixed-species stands highlight the effects of plant species on fungal community structure. Ascomycota, Basidiomycota and Rozellomycota phyla were predominant in all stands, with both Dikarya (Ascomycota and Basidiomycota) accounting for more than 70% in all stands. Correlation analysis revealed that sulfur (S) was the most correlated soil attribute with the three predominant phyla but also with Mucoromycota and Calcarisporiellomycota phyla, although mostly negatively correlated (4 out of 5). Phosphorus was mostly positively correlated to soil attributes (3 out of 4) and nitrogen was correlated twice, positively and negatively. Distance-based redundancy analysis revealed a positive correlation of nitrogen (p-value = 0.0019, contribution = 22%) and phosphorus (p-value = 0.0017, contribution = 19%) with soil mycobiota. A high prevalence of generalists (28% to 38%) than specialists (9% to 24%) were found among the different sites. In stands containing Acacia (pure and mixed species) the soil mycobiota harbor the prevalence of generalist strategies with the potential to withstand environmental stresses and utilize a higher number of resources against specialists in Eucalyptus stands. Stronger positive correlation between soil attributes and main fungal taxa, higher generalists' strategies and lower Faith's phylogenetic diversity and Evenness were reported in stands containing Acacia. This highlights the potential of mixed-species in preserving community stability following environmental disturbances and increasing the number of resources confirming their important ecological role in boosting the resilience of the

(NCBI) "Sequence Read Archive" (SRA) of the National Center for Biotechnology Information (NCBI), under project accession number PRJNA967507.

**Funding:** L.S.K. received funding by the project Ecological Intensification of Plantation Forest Ecosystems (Intens&fix, Agence Nationale de la Recherche, France /Projet-ANR-10-STRA-0004) and by Moët Hennessy, France (LVMH1WLSF2022). Moët Hennessy, France (LVMH1WLSF2022) funded soil analyses (ITS metabarcoding sequencing). A.B. received funding by the European Union's Horizon 2020 research and innovation programme under grant agreement no. 862695 (EJP SOIL) and the Italian project "Creazione di un HUB italiano a support della partecipazione dell'Italia alla Global Soil Partnership ed alla rete di eccellenza europea sulla ricerca sul suolo–SOIL-HUB", granted by the Italian Ministry of Agricultural, Food and Forestry Policies (CUP C52F18000200006). The funders had no role in the design of the study; in the collection, analyses, or interpretation of data; in the writing of the manuscript, or in the decision to publish the results.

forest ecosystems to climate and land-use (plant species as shown by PCA analysis) changes.

## 1. Introduction

Fungi are a vital part of soil microbiomes and perform important functions in forest ecosystems, playing a key role in ecological and biogeochemical processes as decomposers, mutualists, and pathogens [1, 2]. Also, they can act as ecosystem managers by influencing soil structure and physiological processes, and biological regulators, decomposers, and transformers of organic matter [3–7]. Fungi facilitate C sequestration through C translocation into the soil aggregates reducing respiratory activity [8] or low priming effect and high nutrient availability [9]. They sequester C in soil from atmosphere via C absorption in their hyphae enabling symbiosis with plant roots [6]. Due to their diversity and functional contributions, they play a crucial role in forest ecosystems, particularly in the C-cycle, thereby exerting an important influence on the ecological environment [10–12].

Fungi are interconnected with other soil organisms, and changes in fungal activities will affect the entire soil ecosystem [4, 10, 12–14]. Boreal forests harbour the highest levels of soil fungal diversity and richness, while subtropical and tropical forests have the lowest ones. Temperate forests fall in between, showing an intermediate level of diversity [12]. Yet, the fungal community function is crucial in forest ecosystems [4, 10, 12, 14]. It is well known that harvesting and establishing forest plantations affect soil fungi community and composition [4, 15]. In addition to the crucial functions mentioned above, advances in laboratory techniques, the impact of planted forests replacing areas previously used for agriculture and livestock in most cases in Brazil [4, 16, 17], or afforesting (monoculture or mixed-species) native savannas in current case, on fungal communities are receiving increasing attention in the tropics.

Extensive afforestation of native tropical savannas in the Congolese coastal plains has been undertaken, (reaching an overall forested area of around 42.000 ha composed of tropical pines, eucalyptus and acacias in the 2000s), to meet the demand for wood (both in industrial and rural communities), preserve natural forests, mitigate the effects of global warming, and utilize soils unsuitable for agriculture [18]. Amongst them, mixed-species plantations of *Eucalyptus* spp. and *Acacia* spp. were established to improve soil fertility and sustain productivity and meet the growing demand for wood fiber for industry and fuel-wood energy for the local population [19–21]. This practice was found to have a significant impact on below-ground biogeochemical cycling, primarily by inducing changes in the structure of bacterial communities [22]. Moreover, in Brazil, it was observed that intercropping of *Acacia mangium* and Eucalyptus in planted forests significantly increased fungal diversity and richness [4]. Rachid et al. [16] found that intercropping of *Acacia mangium* and Eucalyptus in planted forests significantly increased the number of fungal genera as well as the frequency compared with those found in monoculture plantations. Mixed forest plantations with *Eucalyptus* spp. and *Acacia* spp. regulate the fungal physiological and ecological connections such as fungal richness, diversity, and community complexity and especially the relationship to available P in soil and forest litter [4]. The authors identified forest litter as a vital niche of the fungal community, especially Basidiomycota in boosting P cycling in forest ecosystems. A key role of the forest floor in securing the P cycle (immobilization in organic forms) and reducing its losses via leaching in Acacia and Eucalyptus plantations has been also highlighted [23]. This shows that the interactions between plants, microorganisms, and soil, driven by soil mycobiota, can boost soil functions e.g.,

sustaining the P demand of the ecosystem and enabling C sequestration to further improve soil health.

In a previous study, although the Actinobacteria predominated in all stands, we identified a shift in the bacterial community in stands containing Acacia relative to Eucalyptus with the dominance of *Proteobacteria* in pure *Eucalyptus* against *Firmicutes* in stands containing Acacia [22]. A correlation between the microbial community structure and soil attributes was found, suggesting the important role of soil bacterial community in nutrient cycling in the mixed-species forest plantations. However, the impact of mixed-species plantations on soil mycobiota is still understudied in the forest plantations established in the Congolese coastal plains. This study therefore addresses the hypothesis that mixed-species forest plantations of Acacia and Eucalyptus (monoculture or mixed-species stands) do impact on fungal community as already observed in Brazil [4, 16]. Here, we investigated the impact of mixed-species forest plantations on the soil mycobiota community structure and diversity on the sandy and nutrient-poor soils in the Congolese coastal plains.

## 2. Materials and methods

### 2.1. Ethics statements

The field work was carried out at 35 km from Pointe-Noire in a plateau nearby Tchissoko village (4˚44'41" S & 12˚01'51" E, 100 m a.s.l) in the Republic of Congo. Soil was carried out in the frame of the Intens&fix Project (ANR-2010-STRA-004-03) under the consent of CRDPI (Centre de Recherchesur la Durabilité et la Productivité des Plantations Industrielles) owning the experimental trial. The responsible of the study was Dr. Lydie-Stella Koutika (CRDPI, Local responsible of the project Intens&Fix in Congo). We confirm that our study did not harm the environment and did not involve endangered or protected species.

### 2.2. Site description

The studied site has been previously described in detail [22, 24]. Afforestation using *Eucalyptus* hybrids on native tropical savanna dominated by the $C_4$ *Poaceae Loudetia arundinacea* (Hochst.) Steud. occurred in 1984. Soils are Ferralic Arenosols and have a low content of C, N, and P [25, 26]. The area has a subequatorial climate with an average annual precipitation (1200 mm), a 4 month-long dry season (June to September), and the high mean annual air humidity and temperature (85% and 25˚C, respectively).

Trials (at a density of 800 trees ha$^{-1}$) were installed on a complete randomized block design in May 2004. The trials consisted of three blocks, each composed of three stands; a monoculture stand of *Acacia mangium* (100A), a monoculture stand of *Eucalyptus urophylla* × *E. grandis* (100E), and a mixed-species stand [50% of Acacia and 50% of Eucalyptus (50A50E)]. The first rotation ended in January 2012 (7 years) and the second rotation was established in March 2012 on the same design. As described in Koutika et al. [22], soils were collected with an auger at 0–5 cm, the layer harboring higher soil organic matter (SOM) content, and meso-fauna density and richness at 5 years into the second rotation. A stand comprised 100 trees (10×10) including 36 trees in an inner part on an area of 1250 m$^2$ and two buffer rows. The spacing between rows was 3.75 m, with 3.33 m between the trees in a row. Three transects in 100A and 100E stands and six in 50A50E stands were set up in the inner part of each stand, starting at the base of a tree and ending in the center of the area delimited by four trees. Three cores were sampled on each transect; each sampling point being separated by 0.7 m from the next along each transect. Nine samples were collected in the 100A and 100E stands and 18 in the 50A50E stands with nine samples collected near an Acacia, noted 50A50E (near acacia), and nine others near a Eucalyptus, noted 50A50E (near Euca) in each of the three blocks, as

previously described [22, 24]. The total number of sampling points was 27 (9×3 blocks) in 100A and 100E and 54 (18×3 blocks) in 50A50E. Soil samples were air-dried and sieved at 4 mm. A composite sample was made from three samples of the transect in each stand. Three composite samples of Acacia (100 A) and Eucalyptus (100 E) monoculture and 6 mixed-species (50 A 50 E) stands were obtained by block, i.e., 12 samples per block and a total of 36 samples for 3 studied blocks. Soil samples for chemical analyses have been stored in soil laboratory with constant temperature for one year, whereas samples for molecular analyses were stored at -20˚C in the ENEA C.R. Casaccia (Rome, Italy) until further processing.

## 2.3. Total carbon, nitrogen, sulfur, and available phosphorus analyses

From the 36 collected composite soil samples (3 technical replicates by sample) the above-mentioned analyses have been made using the Macro VARIO Cube Elemental Analyzer (Elementar-Straße 1, D-63505 Langenselbold, Germany) to determine the concentrations of C, N, and S. Available P, also called resin-P, was evaluated using anion exchange resins strips (BDH#551642S, 20 mm × 60 mm). Dried and sieved soils (0.5 g) in 30 mL distilled water and 2 anion exchange resin strips were shaken for 16 h (100 revs min$^{-1}$). After being rinsed with water, the adsorbed phosphate from resin was recovered following elution with 30 mL of 0.5 M HCl. Phosphate was determined according to the method of Tiessen and Moir [27].

## 2.4. DNA extraction and ITS metabarcoding sequencing

The extraction of genomic DNA was performed using the QIAGEN's DNeasy PowerSoil Pro Kit according to the manufacturer's instruction (QIAGEN Group, Hilden, Germany). A total of 36 samples were analyzed. Concentration of extracted DNA was evaluated with a Thermo Scientific™ NanoDrop 2000 spectrophotometer and Qubit 2.0 fluorometer (Invitrogen, Life technologies). The A260/A280 and A260/A230 ratios for DNA samples were > 1.8. The quality and integrity of DNA were checked through 1% agarose gel electrophoresis with 1X Tris Acetate EDTA buffer (Sigma-Aldrich) and GelRed (0.5 μL mL−1) staining. DNA was visualized, and photo-documented under ultraviolet light on Bio Red Molecular Imager ChemiDoc TM XRS+, US. Amplicon libraries of soil mycobiota were generated by amplifying the internal transcribed spacer 1 and 4 using primers ITS1 (5´–TCC GTA GGT GAA CCT TGC GG–3´) and ITS4 (5´–TCC TCC GCT TAT TGA TAT GC–3´) [28] and a subsequent amplification that integrates relevant flow-cell binding domains and unique indices (NexteraXT Index Kit, FC-131-1001/FC-131-1002) using MiSeq Reagent V3 kit (Illumina, Catalog No. MS-102-3003). Sequencing was performed using Illumina MiSeq platform using 300PE v3 chemistry and 300 Paired Ends strategy at IGA Technology Services srl (Udine, Italy) (https://igatechnology.com).

## 2.5. Bioinformatics and statistics analysis

Raw data processing was realized following the DADA2 ITS Pipeline Workflow (1.8) [29]. The taxonomy assignment was performed against the UNITE database (v. 9.0) [30]. Samples were subsequently rarefied at 18,843 reads per sample to normalize read counts across samples. Afterwards, a non-chimeric table of amplicon sequence variants (ASVs) was converted into a 'phyloseq' object [31] into R-studio software (v. 2023.03.0).

To determine the differences in the soil fungal community structure between the Eucalyptus and Acacia species in monoculture and mixed-species, PERMANOVA analysis was performed. Beta-diversity was performed to describe distribution and similarity between sampling groups. Alpha (Pielou evenness and phylogenetic diversity) and beta diversity were calculated, and histograms with the ten and the twenty most abundant phyla and fungi families, respectively, were constructed. To test differences between alpha diversity metrics,

Kruskal-Wallis test was adopted. To investigate the association between the relative abundance of fungi taxonomic groups and soil properties, Spearman's rank correlation coefficients were computed utilizing the 'multtest' package in R. Correction for multiple testing was performed using the Benjamini–Hochberg false discovery rate (FDR). Visualization of the correlations was achieved through the creation of a heatmap using the 'corrplot' package in R, with significant ($P < 0.05$) positive correlations depicted in blue and negative correlations in red. The relationship between soil attributes and fungal communities in different soil samples was also analyzed by distance-based redundancy analysis (db-RDA) [32].

The determination of niche occupancy, denoting the proportion of generalists and specialists within each treatment, was conducted through the multinomial species classification method employing the 'vegan' package in R, specifically utilizing the 'clamtest' function [33]. A significance level of 0.05 and a coverage limit of 10 were set for individual tests. This analytical approach compares the abundance of microbial communities across different environments, categorizing microbes into distinct groups such as specialists, generalists, and those considered too rare. To evaluate the complexity of community interactions, network analysis was carried out. SparCC correlations were computed, with only correlations deemed significant ($p < 0.01$) and strong ($> 0.9$ or $< - 0.9$) being retained [34]. In network analyses each ASV was represented as a node, such that an edge between two nodes implies a relationship between the two corresponding ASVs. The ratio of positive-negative relationships was calculated simply as the number of positively weighted edges divided by the number of negatively weighted edges. Visualization of the networks and calculation of their topological properties were performed using the interactive platform Gephi. A Principal Coordinates Analysis (PCoA) based on the UniFrac distance matrixes was performed to visualize the variations in fungal community structure [35–37].

## 3. Results

### 3.1. Alpha- and beta-diversity

Statistical analysis revealed no significant differences for Faith's phylogenetic diversity in pairwise comparison among the different stands ($p > 0.05$) (Fig 1A and S1 Table). When Pielou Evenness is estimated, alpha-diversity shows higher values in the Eucalyptus monoculture (100E) stands than in stands containing Acacia i.e., monoculture (100A; p = 0,0017), soil samples near Acacia (50A50E near Acacia) (p = 0,0015) and near Eucalyptus (50A50E (near Euca) (p = 0,0009) in mixed-species plantation stands (Fig 1A and S1 Table).

PCA analysis revealed a clear separation of fungal communities in 100E from stands containing Acacia, suggesting the effects of plant species on fungal community structure (Fig 1B). This is confirmed by the statistical analysis reported in S2 Table highlighting a significant difference in fungal beta diversity in mixed-species compared to single stands i.e., (50A50E (near Acacia) vs.100A and 50A50E (near Euca) vs. 100E). Overall, the conversion from monocultures to mixed-species forest plantations affected soil fungal community structure and diversity.

### 3.2. Relative abundance of phyla and families

The relative abundance at the phylum level shows that overall, Ascomycota was dominant (around 60%) followed by Basidiomycota (around 17%) and at a very lower extent by Rozellomycota (2%) (Fig 2A).

Both Dikarya (Ascomycota and Basidiomycota) account for more than 80% in 100A and 50A50E (near Acacia), and more than 70% in 100E and 50A50E (near Euca). A higher prevalence of Ascomycota is observed in 100A (66%), and a lower prevalence in 100E (58%) (Fig 2A). In stands containing Acacia, the relative abundance of Ascomycota ranges between 66%

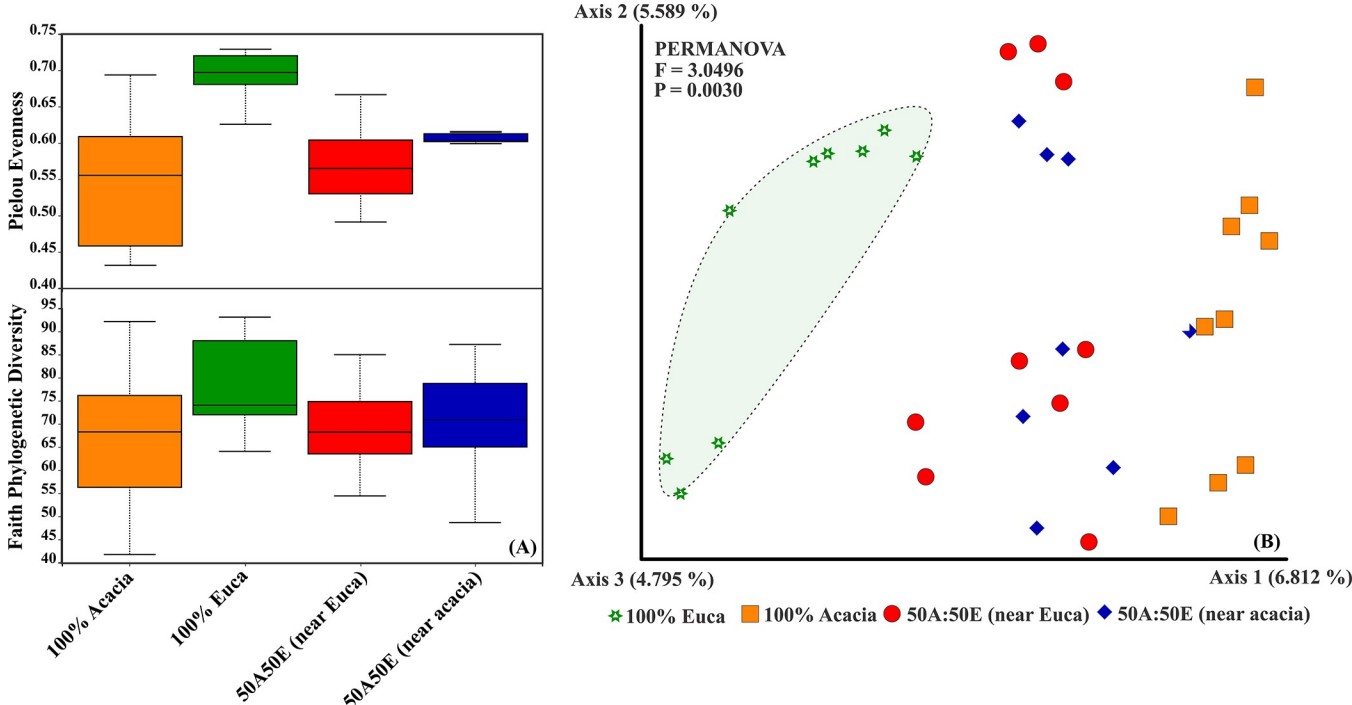

**Fig 1.** Alpha-(A) and Beta-diversity (B) of fungal communities in a sandy soil under mixed-species of Acacia and Eucalyptus plantations at 5 years into the second rotation in the Congolese coastal plains; 100% Acacia = Acacia monoculture stands; 100% Euca = Eucalyptus monoculture stands; 50A50E (near Euca) = soil sampled near Eucalyptus of mixed stands; 50A50E (near Acacia) = soil sampled near Acacia of mixed stands. a) The line inside the box represents the median, while the whiskers represent the lowest and highest values within the 1.5 interquartile range (IQR). The width of the distribution of points was proportionate to the number of points at that *Y* value. b) Principal component analysis (PCA) of fungal communities. Samples are colored according to the plant species and mixed plantations.

(100A) and 62% in [50A50E (near Acacia)], 65% in [50A50E (near Euca)] as intermediate values. The second most dominant phylum, i.e., Basidiomycota represents around 17% of total relative abundance in all stands containing Acacia (monoculture and mixed) and 16% in 100E. The relative abundance of other phyla represents 18% in 100E, 12%, in 50A50E (near Acacia), 10% in 50A50E (near Euca) and 8% in 100A stands (Fig 2A). On the contrary, the lower relative abundance of the unidentified is apparent in 100E (5%) and higher 7% for both 100A and 50A50E (near Acacia) stands.

*Aspergillaceae* is the prevalent family accounting for 56% of the family total relative abundance in 100A, 40% in 50A50E (near Acacia), 34% in 100E and only 28% in 50A50E (near Euca) (Fig 2B). The second most dominant family is *Geminibasidiaceae* covering over 40% of total relative abundance with 48% in 50A50E (near Euca), 43% in 100E, 33% in 50A50E (near Acacia) and only 26% in 100A. *Cortinariaceae* is the third most represented family with 16% in 50A50E (near Euca), 15% in 50A50E (near Acacia) and less than 10% of total relative abundance in both monocultures (6% in 100E and 4% in 100A) (Fig 2B). The family harboring the fourth most prevalent relative abundance is *Lipomycetaceae* with 3% in 100E and 50A50E (near Acacia), 2% 50A50E (near Euca) and only 1% in 100A. Since there were too many unclassified ASVs, further levels were not considered in this study.

### 3.3. Correlation between phyla, families, and soil attributes

Sulfur (S) is the soil attribute showing the greatest number of correlations with phyla; it is negatively correlated with member of the phyla Ascomycota, Rozellomycota, Mucoromycota and

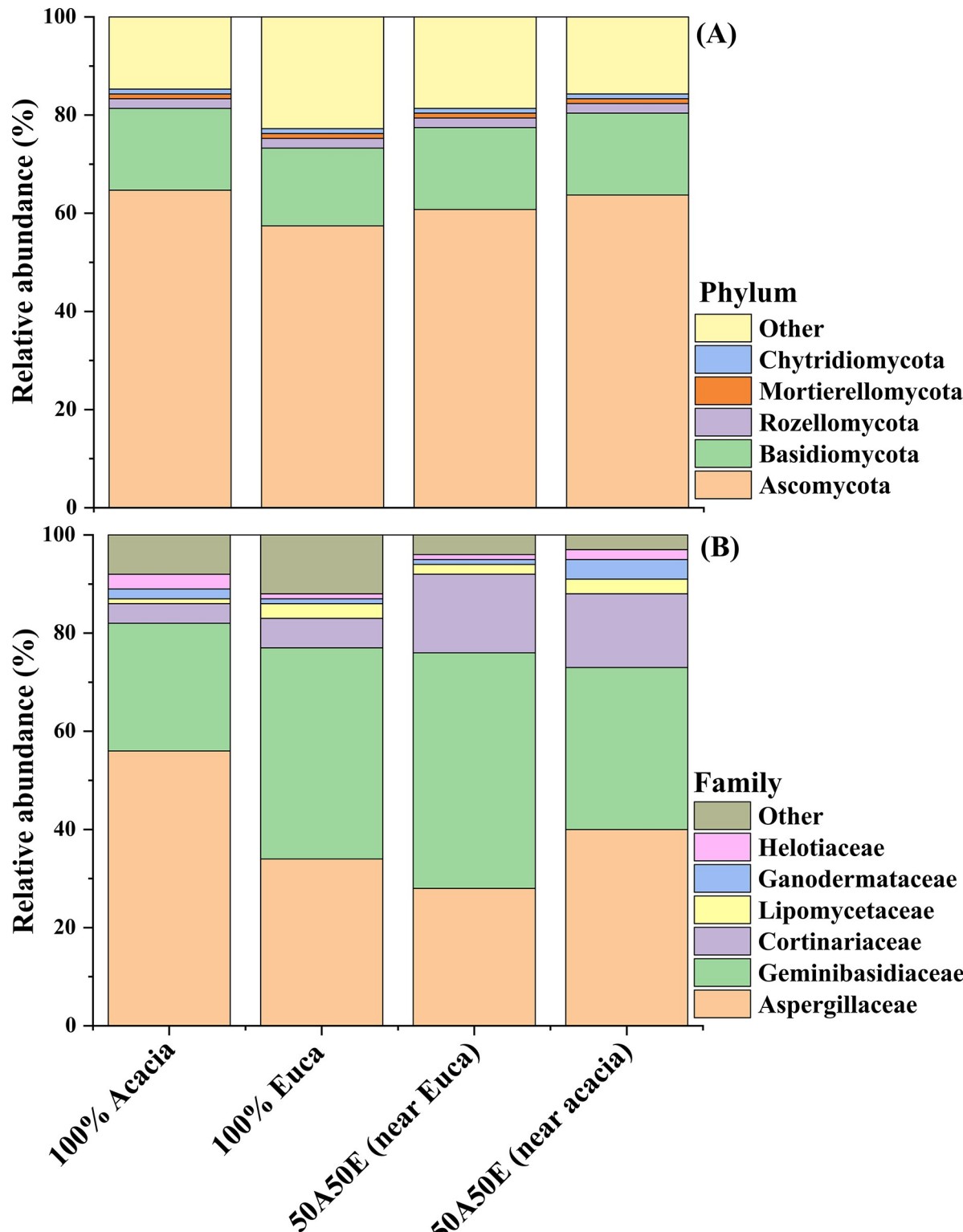

**Fig 2.** Relative abundance of phyla (A) and families (B) in a sandy soil under mixed-species of Acacia and Eucalyptus plantations at 5 years into the second rotation in the Congolese coastal plains. 100% Acacia = Acacia monoculture stands; 100% Euca = Eucalyptus monoculture stands; 50A50E (near Acacia) = soil sampled near Acacia in mixed plantations; 50A50E (near Euca) = soil sampled near Eucalyptus in mixed plantations. Other: unclassified taxa and other fungal phyla with low amplicon sequence variants (ASVs) abundance.

Calcarisporiellomycota and positively correlated with Basidiomycota ([Fig 3A]). Phosphorus (P) is positively correlated with Mortierellomycota, Mucoromycota and Glomeromycota and negatively with Basidiomycota. Nitrogen (N) is negatively correlated with Glomeromycota and positively correlated with Entomophthoromycota. No correlation is found for carbon (C) at phylum level ([Fig 3A]). Ascomycota, Rozellomycota and Mucoromycota are positively correlated with C/S, while Chytridiomycota is also positively correlated with C/N.

Spearman scores show that P is the soil attribute that is linked seven families, i.e., positively to five (*Geminibasidiaceae*, *Hydnodontaceae*, *Sporormiaceae*, *Archaeeorhizomycetaceae* and *Mortierellaceae*, and negatively to two (*Thermoascaceae*, *Ganodermataceae*) ([Fig 3B]). Six families are correlated with N, i.e., positively (*Helotiaceae*, *Myxotrichaceae* and *Testudinaceae*) and negatively (*Botryosphaeriaceae. Thermoascaceae*, *Nectriaceae*). Both C and S are each correlated with only 3 families. Carbon is correlated positively with *Aspergillaceae* and *Myxotrichaceae*, and negatively with *Trimorphomycetaceae*. S is positively correlated with *Lipomycetaceae*, *Microascaceae* and *Sympoventuriaceae* ([Fig 3B]). Despite correlation found between other soil attributes and ratio (C, S, C/N and C/S), according to the distance-based redundancy analysis (db-RDA), only nitrogen (p-value = 0.0019, contribution = 22%) and phosphorus (p-value = 0.0017, contribution = 19%) were the soil properties contributing more significantly to changes in fungal community structure and diversity ([Fig 4]).

### 3.4. Niche occupancy and networks

The microbial niche occupancy analysis revealed a higher proportion of generalists (28%-38%) respect to specialists (9%-24%). A range of 28% to 35% of fungi are rare. More importantly, Eucalyptus monoculture plantations (100% Euca) showed a higher number of specialists (24,7% of fungal ASVs) than Acacia stands alone (15.6%) (P<0.05). A few specialists dominated fungal communities of the soils in mixed stand plantations; i.e., 12.5% in 50A50E (near Euca) and 9% in 50A50E (near Acacia) ([Fig 5A]). When comparing 100% Euca and mixed-species plantations (near Acacia), an increased proportion of generalists (38%) was found while decreasing the proportion of specialist microbes in 50A50E near Acacia ([Fig 5A]).

The fungal ecological network analysis revealed differences when comparing mixed-species with the respect to monoculture stands. An extensively interconnected fungal community was found in 100% Euca, presenting a higher number of nodes (61), edges (140), positive (107) and negative (34) correlations compared to the other sites ([Fig 5B]).

## 4. Discussion

The soil mycobiota is considered as a key factor in the ecological functions of forests through the provision of resources for plant nutrition and productivity [38]. Nitrogen-fixing species (*Acacia mangium*) introduced to sustain Eucalyptus plantations and improve soil fertility did affect soil mycobiota community structure and diversity of the mixed-species forest plantations. The current study aimed to shed light on the role of fungi in driving soil functions and fertility and fostering adaptation and resilience to land-use and climate changes.

### 4.1. Effect of mixed-species plantations on soil fungal community structure and diversity

Differences in alpha and beta diversity were found among Eucalyptus and Acacia monoculture, and mixed-species stands. A higher Faith's phylogenetic diversity, which reflects the sum of all branch lengths on the constructed phylogenetic tree from all taxa (alpha-diversity), was found in Eucalyptus monoculture compared to stands containing Acacia. In addition, Eucalyptus monoculture harbors a distinct fungal community as shown by beta-diversity analysis,

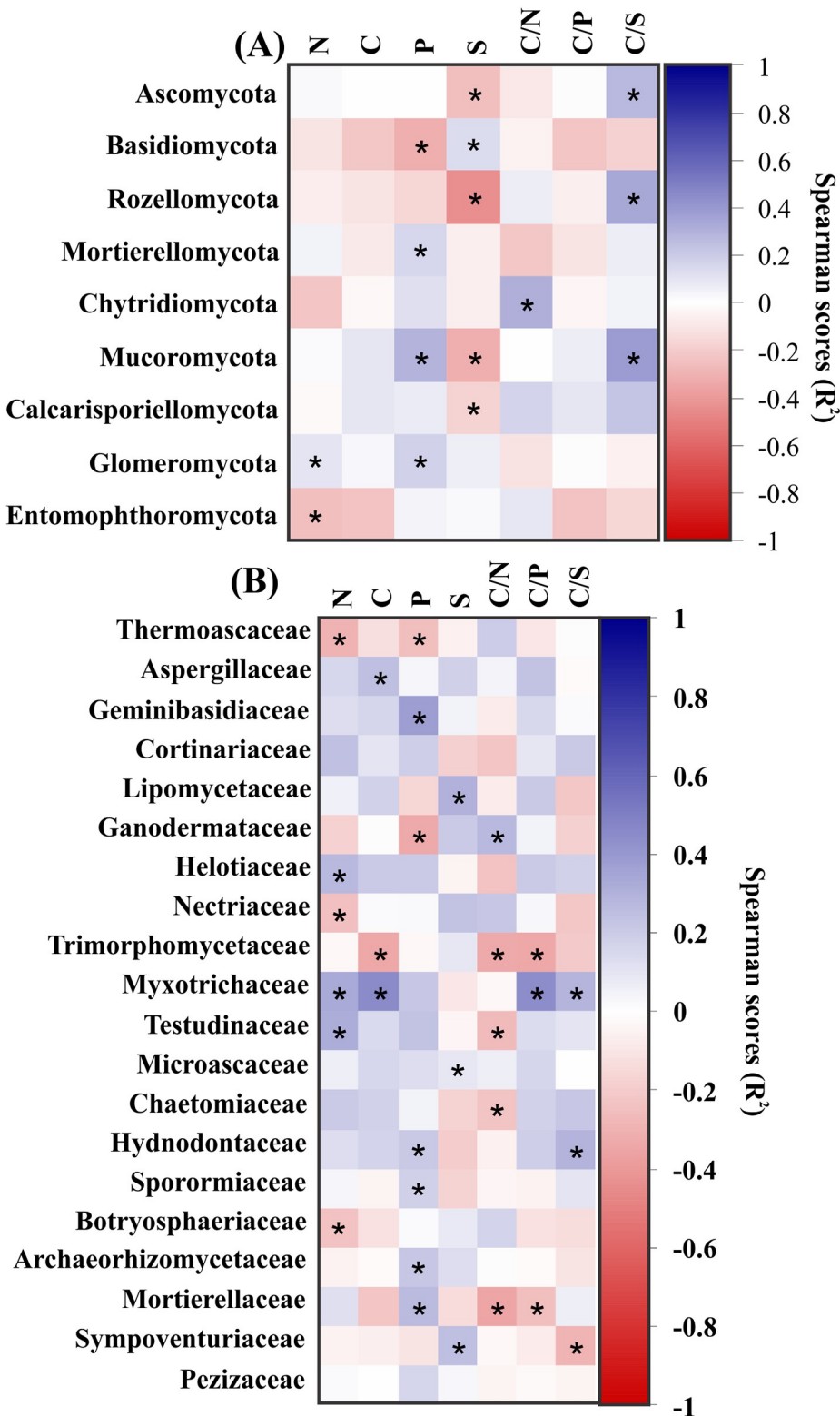

**Fig 3.** Correlation matrix of Spearman scores between soil attributes and dominant phyla (A) and families (B) that are significantly (P = 5%) correlated, either positively or negatively. The values of correlation coefficients are indicated according to the scale bar. Positive correlations are shown in blue while negative correlations are shown in red. The asterisks represent a significant correlation.

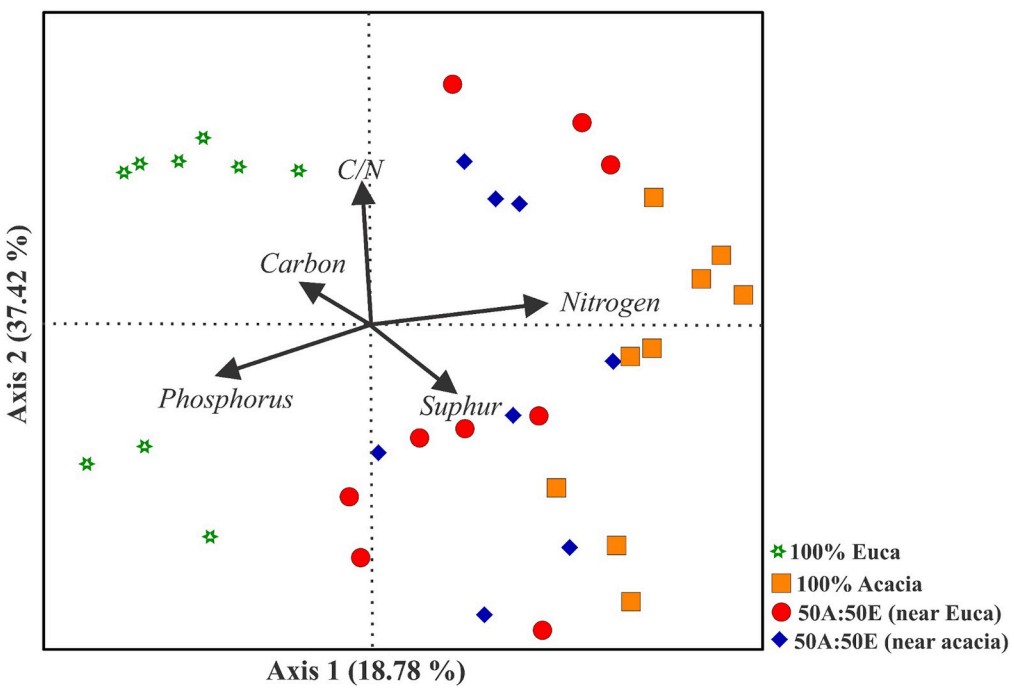

**Fig 4. Correlation between soil chemical attributes (phosphorus, C/N, nitrogen, sulfur, and carbon) and soil mycobiota composition by distance-based Redundancy Analysis (db-RDA).**

which revealed more negative correlations and negative edges. Our result is in accordance with Rachid et al. [16] who reported an enhanced number of fungal genera and their frequency in monoculture plantations in intercropped *Acacia mangium* and *Eucalyptus* plots. It is well known that microbial generalist and specialists showed a different contribution to the community diversity [39]. Generalist fungi are characterized by stochastic processes, higher diversification and transition rates enabling them to play an important role in preserving community stability following environmental disturbances [39]. They also shape the fungal community structure and are adaptable to various environmental conditions [40]. On the contrary, specialists are expected to contribute more to deterministic processes (e.g., biotic and abiotic environmental filtering) and are more sensitive to environmental variations and support the soil microbial ecosystem functions [41]. Our results revealed a high prevalence of generalists (28% to 38%) than specialists (9% to 24%) among the different sites. Niche occupancy analysis revealed that mixed-species plantations favored generalists over specialists. Our findings suggest that mixed-species plantations potentially increase the ability of soil mycobiota to withstand environmental stress, favoring its pivotal role in enhancing soil ecological stability [39, 42]. This finding may also be related to the higher ratio of positive to negative edges reported in mixed-species plantation near Acacia (53/12 = 4,42), followed by those in Acacia monoculture (76/18 = 4.22), whereas the ratio was 107/34 = 3,14 in Eucalyptus monoculture and the lowest in mixed plantation near Eucalyptus (71/26 = 2.73). It is well known that the positive correlation could indicate cooperative interactions or the presence of common biological functions or ecological niche between taxa. On the contrary, negative correlations could be indicative of either competitive interactions or non-overlapping ecological niches or processes between taxa. The highest percentage of specialists in Eucalyptus monoculture suggests that this condition preserves more keystone species and their connectivity, as revealed by the greater number of nodes in the site under the former vegetation; however, these nodes

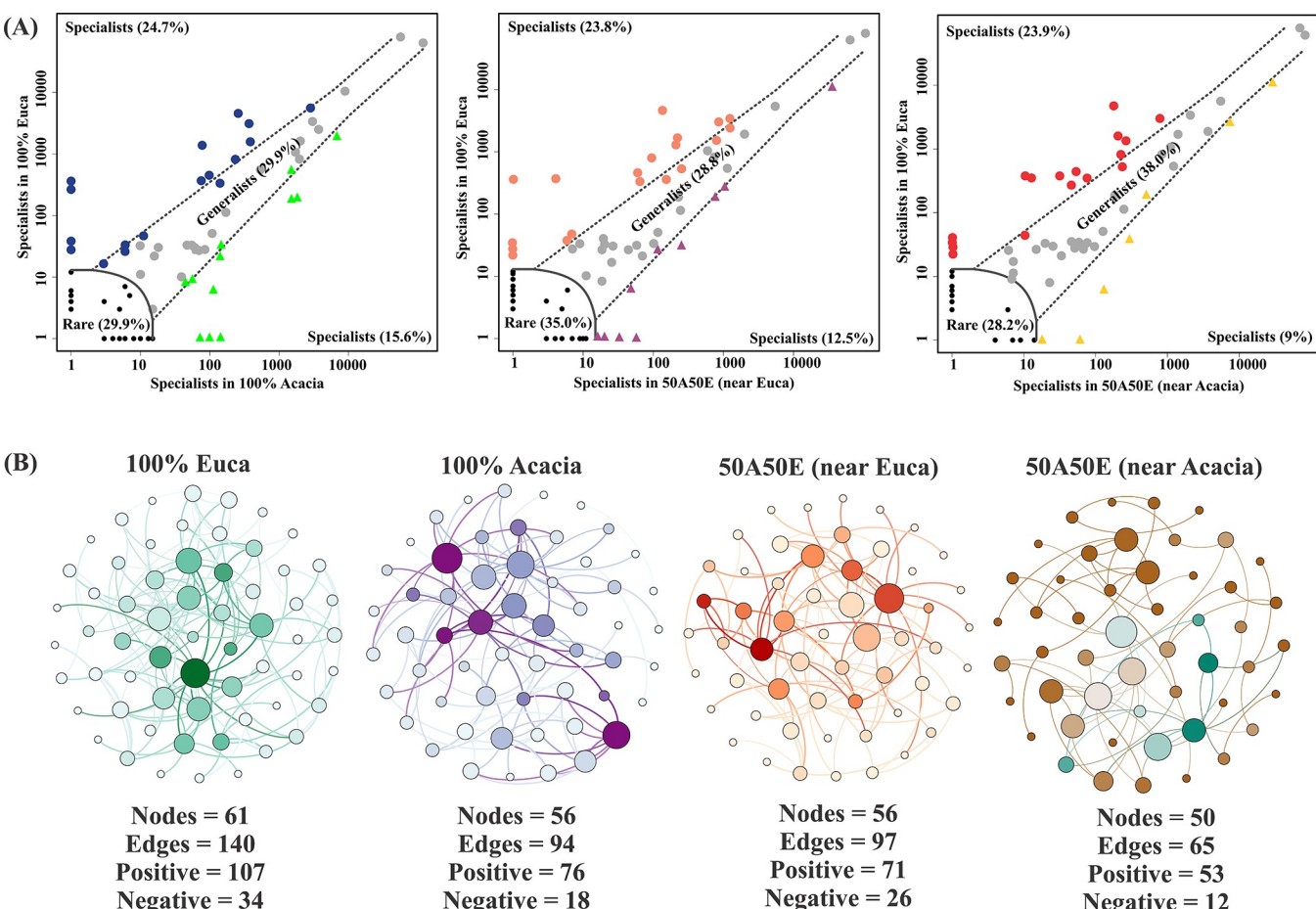

**Fig 5.** Niche occupancy (A) and networks (B) in sandy soil under mixed species of Acacia and Eucalyptus plantations at 5 years into the second rotation. 100% Acacia = Acacia monoculture stands; 100% Euca = Eucalyptus monoculture stands; 50A50E (near Acacia) = soil samples near Acacia (50A50E (near Acacia) in mixed plantations; 50A50E (near Euca) = soil samples near Eucalyptus in mixed plantation; Black and grey: rare and generalists, respectively. Blue, Orange, and Red: specialists in 100% Euca Green: specialists in 100% Acacia Purple: specialists in 50%A50%E near Euca Yellow: specialists in 50%A50%E near Acacia.

displayed higher negative connections. This outcome implies potential competition between keystone species, which could potentially compromise the functionality of the soil [43]. In addition, decomposition rate of Eucalyptus leaf litter poor in nitrogen and lignin is lower that of Acacia containing high content of nitrogen and water-soluble carbon [44]. This may involve the formation of a more diversified fungal community distinct in Eucalyptus relative to those in stands containing Acacia. Fungi are the main decomposers of plant material, primarily the lignocellulosic components, which are relatively recalcitrant to bacteria [45], as they produce oxidative enzymes enabling the decomposition of the recalcitrant biopolymers of plant litter [46, 47].

Overall, fungal communities in stands containing Acacia relative to Eucalyptus harbor distinct fungal communities and lower Faith's phylogenetic diversity. In a study conducted on the same samples, bacterial communities were found to be distinct in Eucalyptus stands from those of stands containing Acacia through the higher bacterial richness and Faith's phylogenetic diversity [22]. The current study reports even a more connected fungal community structure in Eucalyptus monoculture reflecting stability in microbial communities [48, 49]. This is in line with our hypothesis which highlights the effect of stand type (monoculture or mixed-

species stands) on fungal community structure and diversity. Although it is not clear whether richness of host plants affects local ectomycorrhizal fungi species richness as shown elsewhere [43], it is probably related to different factors, such as soil properties, plant distribution and dispersion capacity and size of the fungal species inhabiting the environment. Species richness does not explain by itself the richness variation of ectomycorrhizal fungi in complex environments, but there is a strong effect plant species in shaping ectomycorrhizal fungi community, defined as the 'host effect'. This effect could be more intense in sites with low plant diversity [50].

Relative abundance of Ascomycota was higher in stands containing Acacia relative to Eucalyptus. This finding is in line with Rachid et al. [16] who found that that the plant type had a major effect on the soil fungal community structure in the forestry system. The prevalence of Dikarya (Ascomycota and Basidiomycota) in all examined stands i.e., over 70% are in accordance with other studies (4,12). Ascomycota and Basidiomycota are commonly the two dominant phyla in tropical forests (up to half of the fungal community) [51] and temperate forest [1] ecosystems, or for instance in tea plantations converted from forest [40]. The occurrence of Ascomycota in the studied afforested stands in the Congolese coastal plains may be due to the ability of this phylum to degrade organic substrates such as wood, leaf litter, etc. [52, 53]. Egidi et al. [52] reported patterns and ecological drivers of dominant soil fungal taxa occurrence worldwide. Ascomycota taxa dominate the soil fungal community because they possess greater potential to adaptation to various environmental conditions [52]. Mayer et al. [54] stated that although Ascomycota taxa are less efficient in decomposing SOM than Basidiomycota, they produce hydrolyzing enzymes which decompose SOM. A higher relative abundance of Ascomycota in more fertile plots with enhanced SOM decomposition was observed [54]. The prevalence of Ascomycota (mainly ectomycorrhizae) in stands containing Acacia may also be related to high C storage [55], N status [56] and higher available P in the forest floor [23], previously reported in this site. Acacia and Eucalyptus plantations support a particular niche of fungal communities which encourage and stimulate release of available P in the litter layer, thereby enhancing the relationship between fungi and nutrient cycling [4, 12]. *Aspergillaceae*, an ectomycorrhizal fungi family, is dominant in all stands, while the *Geminibasidiaceae* family has higher relative abundance in stands containing Acacia (Fig 2B). Mixed-species plantations often harbor different functional fungi which are involved in soil nutrient cycling [57]. This may be linked to changes reported in soil attributes (C, N, P) in the current studied Acacia and Eucalyptus plantations [24, 25, 56].

## 4.2. How do forest plantations impact fungal communities?

Anthropogenic activities such as land-use change (afforestation) may have induced swift changes in the bacterial and fungal communities [4, 58]. Fungi can enhance soil organic carbon (SOC) dynamics (C storage) and nutrient cycling [4, 5, 8, 9, 59]. Fungi are widely acknowledged as more effective decomposers of recalcitrant organic materials compared to bacteria, owing to their broad enzymatic capabilities and specialized metabolic pathways [44]. Although no significant correlation between Ascomycota and C was found study at phylum level, we found a strong positive correlation between the total C, C/P and C/S of the soil and the Myxotrichaceae family. Saprophytic ascomycetes belonging to this family has been isolated from decaying logs in the boreal forest [60] and are able to degrade cellulose and tannic acid having the potential to cause significant mineralization of carbon in bogs [61]. Studies conducted on bacterial communities revealed only four Spearman correlations (at phylum level) i.e., two between S and *Actinobacteria* and *Chloroflexi* and another two between C and *Chloroflexi* and *Planctomycetes* [22]. The link between fungi and soil attributes is more evident

confirming fungi as key players in boosting soil fertility e.g. by their role in the C-cycle, but also in driving the relationship between soil fertility, ecological environment, humans, and human activities [10, 12, 36].

Spearman correlations revealed that S is the soil attribute showing the greatest number of correlations (five) between fungi and phyla including Ascomycota, and Mycoromycota, although mainly negatively (4 out of 5). Phosphorus which showed correlations with four phyla, mainly positively (3 out of 4), including Mycoromycota, ranked second. Galitskaya et al. [62] reported a high prevalence of Mycoromycota, and a continuous increase in Ascomycota against a decrease in Basiomycota abundance in a 30-day experiment of oil-polluted soils. At the family level, P was ranked first with seven correlations (5 positive out of 7) while S had only three although all positive. Fungal species able to degrade lignin, hemicellulose and cellulose may dominate thanks to their nonspecific enzymes enabling aromatic compounds degradation, as suggested by Galitskaya et al. [62]. This study highlighted the influence of mixed-species forest plantations in the Congolese coastal plains on soil fungal community structure and diversity and their relation to soil attributes. The impact on fungi is important as highlighted by a stronger correlation with soil attributes as shown by correlations for S (4 negative out of 5), P (3 positive out of 4), and N (one positive and one negative).

Overall, our results underscore the interplay between fungi-driven biotic and abiotic components of soil habitat, as well as the potential influence of human activities on soil attributes and the environment. Mixed-species stands seem to engender soil mycobiota which enhance the potential to withstand environmental stress, involving greater link to soil attributes, mainly S, although mainly negatively, P and N than bacterial communities. Enhanced N status was the main benefit of introducing nitrogen-fixing species in Eucalyptus plantations. Nitrogen is closely linked to P which is needed for the symbiotic fixation of atmospheric nitrogen. Phosphorus is strongly correlated to fungi and mainly positively. It is also the second most correlated soil attribute at the phylum level after S and first at family level. Sulfur (S) is the soil attribute harboring a higher correlation at phylum level including Ascomycota, the prevalent phyla, and Mycoromycota, negatively correlated to both. The potential impact of the above-mentioned anthropogenetic activities on soil mycobiota and environment needs to be further investigated as well as their implications for ecosystem health and sustainability. Studies in taxonomic and functional composition of soil mycobiota and microbiomes, and their link to soil fertility and environment, should be conducted in other areas of the country.

## Supporting information

**S1 Table. Faith's phylogenetic diversity and Pielou Evenness p-values obtained from of Kruskal-Wallis test of fungal communities in a sandy soil under mixed-species of Acacia and Eucalyptus plantations at 5 years into the second rotation in the Congolese coastal plains.**
(DOCX)

**S2 Table. Pairwise Permanova results of fungal communities in a sandy soil under mixed-species of Acacia and Eucalyptus plantations at 5 years into the second rotation in the Congolese coastal plains.**
(DOCX)

## Acknowledgments

The authors warmly thank the staff of CRDPI, Pointe-Noire, Republic of the Congo for their contribution during the establishment of the experimental trial and soil sampling, G. Aprea

for the submission to Sequence Read Archive (SRA) of high throughput sequencing data, and M. Iannetta (Head of the ENEA "Sustainable AgriFood Systems"- Division) and the colleagues of the project SUS-MIRRI.IT (https://www.sus-mirri.it) for helpful discussion.

## Author Contributions

**Conceptualization:** Lydie-Stella Koutika, Arthur Prudêncio de Araujo Pereira, Annamaria Bevivino.

**Formal analysis:** Arthur Prudêncio de Araujo Pereira, Alessia Fiore, Annamaria Bevivino.

**Funding acquisition:** Lydie-Stella Koutika, Annamaria Bevivino.

**Investigation:** Lydie-Stella Koutika, Arthur Prudêncio de Araujo Pereira, Silvia Tabacchioni, Manuela Costanzo, Luciana Di Gregorio, Annamaria Bevivino.

**Methodology:** Arthur Prudêncio de Araujo Pereira, Alessia Fiore, Silvia Tabacchioni, Manuela Costanzo, Luciana Di Gregorio.

**Project administration:** Lydie-Stella Koutika, Arthur Prudêncio de Araujo Pereira.

**Resources:** Lydie-Stella Koutika, Annamaria Bevivino.

**Software:** Arthur Prudêncio de Araujo Pereira.

**Supervision:** Lydie-Stella Koutika, Annamaria Bevivino.

**Validation:** Lydie-Stella Koutika, Alessia Fiore, Silvia Tabacchioni, Manuela Costanzo, Luciana Di Gregorio, Annamaria Bevivino.

**Visualization:** Lydie-Stella Koutika, Arthur Prudêncio de Araujo Pereira, Silvia Tabacchioni, Manuela Costanzo, Luciana Di Gregorio, Annamaria Bevivino.

**Writing – original draft:** Lydie-Stella Koutika, Arthur Prudêncio de Araujo Pereira, Annamaria Bevivino.

**Writing – review & editing:** Lydie-Stella Koutika, Arthur Prudêncio de Araujo Pereira, Alessia Fiore, Silvia Tabacchioni, Manuela Costanzo, Luciana Di Gregorio, Annamaria Bevivino.

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
