## [Decision Letter · Decision Letter 0]

24 Jun 2024

PONE-D-24-15839Impact of mixed-species forest plantations on soil mycobiota composition and diversity in the Congolese coastal plainsPLOS ONE

Dear Dr. Bevivino,

Thank you for submitting your manuscript to PLOS ONE. After careful consideration, we feel that it has merit but does not fully meet PLOS ONE’s publication criteria as it currently stands. Therefore, we invite you to submit a revised version of the manuscript that addresses the points raised during the review process.

We look forward to receiving your revised manuscript.

Kind regards,

Tunira Bhadauria, Ph.D.

Academic Editor

PLOS ONE

Journal Requirements:

Reviewers' comments:

Reviewer's Responses to Questions

**Comments to the Author**

1. Is the manuscript technically sound, and do the data support the conclusions?

Reviewer #1: Yes

Reviewer #2: Yes

2. Has the statistical analysis been performed appropriately and rigorously? 

Reviewer #1: I Don't Know

Reviewer #2: Yes

3. Have the authors made all data underlying the findings in their manuscript fully available?

Reviewer #1: Yes

Reviewer #2: Yes

4. Is the manuscript presented in an intelligible fashion and written in standard English?

Reviewer #1: Yes

Reviewer #2: Yes

5. Review Comments to the Author

Reviewer #1: # Reviewer

The manuscript deals with the investigation of the fungal community in pure and mixed plantations of eucalypt and acacia in the Congo. The information is relevant and unprecedented for the region. The information was analyzed by different statistical techniques and is presented in figures. However, my perception is that the manuscript needs a broad revision to simplify the message and direct the discussion to the results of the present study. In addition, the authors cite different articles developed apparently in the same experimental device, but the relationship between the results is not very evident in the discussion.

L17-18. Include the full scientific name of the tree species.

L 29-33: It's confusing. Which planting model(s) provide the most benefits?

L 33-35 This phrase makes more sense at the beginning of the results (L19).

L 35-37. It's confusing. Soil attributes were also related to monospecific plantations. It is unclear what positive effect it has on the resilience of the ecosystem.

L47. It could include citations for this statement or specify which regions this occurs. My perception is that interest in mycobiota has increased due to advances in laboratory techniques. The interest is not evident in the conversion of natural forests to planted forests in all tropical regions. In Brazil, for example, planted forests replace areas previously used for agriculture and livestock in most cases. One interest that seems to be growing is to understand the feedback between plant diversity (e.g., by converting pure forest plantations to mixed plantations) and microbial communities. Perhaps mentioning something about this would help link to the next paragraph.

L 54. Why Eucalyptus and/or Acacia? It needs a minimum of two species.

L 71-74. My perception is that this sentence is confusing and perhaps in the wrong place. Was this territorial extension occupied only by mixed plantations of eucalypt and acacia?

L 79. What type of forest management?

L80-81. Could you specify which forest management and type of stand the authors are comparing? Could you include references to work done elsewhere to support your hypothesis?

L 82. Don't composition and diversity have the same meaning in the sentence? Wouldn't it be more appropriate to replace it with community structure and diversity?

L 89-91. Average air temperature is an important piece of information as well.

L 92-93. The first sentence is more in line with the first paragraph.

L94. The description of the experimental device seems confusing to me. Are there three replications with monospecific acacia plantations? And just one stand with eucalypt monoculture? Is it just a stand with mixed planting? Would this be a complete randomized block design?

L 98-100. Wasn't there a new mixed planting? It's confusing because it seems that there is no mixed planting for data collection.

L 105. What is the length of the transect? 10 m (similar to the width of the sampling plot)? If that's the case, it seems odd that the sampling points are only 0.7 m apart from each other.

L110-115. Routine soil fertility practice is to use a 2 mm sieve. Also, it seems confusing to me the way he mixed the simple samples to form the composite samples. In addition, it is unclear how long the sampling and molecular analysis took place, and how the samples were transported from the field to the lab.

L117. The elemental analyzer provides the total content of each element. So, it would be appropriate to include the term Total (e.g. Total C, N, and S...).

L129-130. Specify how these samples were distributed among treatments.

L 152. Wouldn't a parametric test (ANOVA, Tuckey, etc.) be more appropriate since the data come from a randomized experimental device? Ditto for Spearman's correlation, which could be replaced by Pearson. Has the normality of the data been tested?

L 186-189. I don't remember reading that described in the methodology that the samples were collected near eucalypt or acacia trees in mixed plantations.

L 189-192. Both sentences should be in the methodology.

L 192-196. It seems to me that the association of several statistical techniques to present the results is confusing. I miss a greater link between the first sentence (PCA) and the second sentence (Table S2).

L 217. Report this proportion of unidentified phyla.

L231. Authors should avoid single-sentence paragraphs. In addition, the S does not seem to have been the most important element in explaining the differences between the treatments. Perhaps it would be better to separate the information into positive and negative correlations, as this leads to different impacts.

L252-254. Does this mean that only P and N contributed significantly? C and S, or C/N, C/P, N/P... Have they not contributed significantly?

L263. Have any tests been conducted to assess this? Was the difference statistically significant?

L 265. The legend of this figure should be improved. There are color variations and colors that have no meaning (such as the shades of red, and orange in 50E:50A near euca).

L271-272. The authors could more adequately explain these results, given the size of the figure, and the different patterns colors, and connections. How are these connections made?

L 276-284. This paragraph is more appropriate in the introduction than in the discussion.

L293-294. The difference between generalists and rare ones was small. The sum of the two can represent up to 73% of mycobiota. One question would be whether specialist species have a particular ecological function.

L 296. Eucalyptus in this case should be italicized.

L296-297. That sentence seems confusing to me. If the number of species is greater in pure plantations, wouldn't they be more resilient to environmental stresses? Would increasing the diversity of fungi in theory be more beneficial, or not?

L297-300. The authors could better explain these ratios since they were not presented at the M&M.

L 300-306. I have doubts whether the eucalypt litter would necessarily be more recalcitrant than that of the acacia. The crucial difference seems to me to be the N content, which is higher in acacia leaves. But acacia leaves take longer to decompose than eucalypt leaves.

L 311. “.. potential associations with function (40,41)”. Quais functions?

L 311-313. Although there are no studies on the diversity of the fungal community in forest plantations of eucalyptus, acacia, and the mixture of the two species in the Congo, this hypothesis is somewhat presumptive to be accepted. The authors cite studies showing that microbial communities are to some extent modulated by the species planted on the site (e.g. 4, 17, 18). Perhaps the authors could go beyond that. For example, the authors mention in the introduction that fungi are important for carbon sequestration in the ecosystem, forest productivity, etc. Wouldn't it be possible to make that kind of relationship? For example, the abstract states: "Higher Faith's phylogenetic diversity and Evenness' was found in Eucalyptus monoculture relative to stands containing Acacia." Did this result in improved or resistance in some productivity parameters (e.g. growth, tree biomass, survival, litterfall?

L 314. I don't recall reading a highlight about endomycorrhizae versus ectomycorrhizae in M&M and outcomes. I think it's important.

L 360-364. The authors highlight important functions of fungi in ecosystems in these sentences but have little to do with the results. For example, it could explain the strong positive correlation between the total C (and C/P) of the soil and the Myxotrichaceae family.

L322-323. This statement is confusing. What the authors mean by "may have been compensated." Is it related to the growth/survival of trees?

L 335-336. The authors could include some explanation for the lack of correlation of this phylum (Ascomycota) with the total C contents.

L337-350. Were the percentage differences statistically significant? In Figure 2A, there do not seem to be major differences, but %100E and %100A show extreme values, while mixed plantings show intermediate percentages for some phyla and families. Perhaps it is more appropriate to concentrate the discussion on the phyla and families that were correlated with soil fertility parameters.

L 356-358. It is unclear what impact these issues (afforestation and oil production) will have. What would be the potential impact of this? (e.g. for productivity, ecosystem services of plantations?).

L360-361. This statement would make more sense if information about fungal communities in the reference area as native ecosystems were presented.

L369-371. Authors could include references to support this claim. It seems to me inappropriate to assume this based on correlations.

L 379-382. This statement seems limited to me. What would be the anthropic activity? Wouldn't it be better to discuss it from the perspective of diversity and its effects?

L382-383. How strong was this correlation? Was it always positive or negative? It seems to me that there is a lack of discussion in this regard.

L383-386. My perception is that there is a lack of a more evident link between human activities and soil alteration. In addition, the last sentence does not seem to be consistent with Figure 3.

L387-399. That paragraph is a conclusion, but the information could be better discussed in the discussion.

Reviewer #2: Dear Authors, the submitted manuscript has been written well and scientifically sound. I did not find any scientific mistakes in the manuscript hypothesis and research program. After careful review, I would like to suggest for some minor corrections: The abstract section could be more informative and precise with data as per the experiments performed and results obtained.

The introduction parts seem good, hence, I have no suggestion to improve this part. In the material method section, the first heading contains details that are not required, therefore, it could be improved by writing more in fewer sentences. The result section has been written well based on the findings, however, the findings could be improved by mentioning the concerned activity. The general readers could not understand which inferences are concerned with which activity. A major revision needs to be done in the discussion part. It's too lengthy leading to loss of connectivity. The discussion should be crisp and in a concise manner with supporting references.

6. PLOS authors have the option to publish the peer review history of their article (what does this mean?). If published, this will include your full peer review and any attached files.

Reviewer #1: No

Reviewer #2: **Yes: **Deepanshu Jayaswal

---

## [Author Response · Author response to Decision Letter 0]

30 Jul 2024

Dear Editor,

We are pleased to submit the revised version of our manuscript PONE-D-24-15839 for publication in PLOSONE Journal. We would like to express our appreciation to you and the Reviewers for the constructive comments and suggestion, which proved insights into improving our manuscript. 

The manuscript has been thoroughly revised accordingly to your and reviewers’ suggestions. All changes made in the article are tracked in blue colour to facilitate the review process. 

Below, you can find a detailed response letter addressing each reviewers' comments point-by-point.

We hope that the amended version of our paper now meets your final approval. Thank you for the opportunity to revise our work, and we look forward to hearing from you at your earliest convenience.

Yours sincerely,

Annamaria Bevivino and Stella Koutika

5. Review Comments to the Author

Reviewer #1#: 

Reviewer 1’s comment 1: 

The manuscript deals with the investigation of the fungal community in pure and mixed plantations of eucalypt and acacia in the Congo. The information is relevant and unprecedented for the region. The information was analyzed by different statistical techniques and is presented in figures. However, my perception is that the manuscript needs a broad revision to simplify the message and direct the discussion to the results of the present study. In addition, the authors cite different articles developed apparently in the same experimental device, but the relationship between the results is not very evident in the discussion.

Response to reviewer 1’s comment 1: 

We thank the reviewer for valuable comments and suggestions. As suggested, we made a substantial revision especially in the Discussion section. Different aspects highlighted by the reviewers have been considered and developed further, and the relationship between different articles from the same experimental device and the current findings has been carefully considered.

Reviewer 1’s comment 2: 

L17-18. Include the full scientific name of the tree species.

Response to reviewer 1’s comment 2: 

Lines 25-26: The full Latin names of three species (i.e, Acacia mangium, Eucalyptus urophilla, E. grandi) have been reported.

Reviewer 1’s comment 3: 

L 29-33: It's confusing. Which planting model(s) provide the most benefits?

Response to reviewer 1’s comment 3: 

We thank the reviewer for the valuable comment. The sentence has been rewritten:

Lines 38-42: “A high prevalence of generalists (28% to 38 %) than specialists (9% to 24%) were found among the different sites. In stands containing Acacia (pure and mixed species) the soil mycobiota harbor the prevalence of generalist strategies with the potential to withstand environmental stresses and utilize a higher number of resources against specialists in Eucalyptus stands”.

Reviewer 1’s comment 4: 

L 33-35 This phrase makes more sense at the beginning of the results (L19).

Response to reviewer 1’s comment 4: 

The sentence was move to results section. 

Reviewer 1’s comment 5: 

L 35-37. It's confusing. Soil attributes were also related to monospecific plantations. It is unclear what positive effect it has on the resilience of the ecosystem.

Response to reviewer 1’s comment 5: 

We thank the reviewer for this important comment. We therefore reformulated and strengthened the meaning of the sentence. 

Lines 42-54: “Stronger positive correlation between soil attributes and main fungal taxa, higher generalists’ strategies and lower Faith’s phylogenetic diversity and Evenness were reported in stands containing Acacia. This highlights the potential of mixed-species in preserving community stability following environmental disturbances and increasing the number of resources confirming their important ecological role in boosting the resilience of the forest ecosystems to climate and land-use (plant species as shown by PCA analysis) changes.

Reviewer 1’s comment 6: 

L47. It could include citations for this statement or specify which regions this occurs. My perception is that interest in mycobiota has increased due to advances in laboratory techniques. The interest is not evident in the conversion of natural forests to planted forests in all tropical regions. In Brazil, for example, planted forests replace areas previously used for agriculture and livestock in most cases. One interest that seems to be growing is to understand the feedback between plant diversity (e.g., by converting pure forest plantations to mixed plantations) and microbial communities. Perhaps mentioning something about this would help link to the next paragraph.

Response to reviewer 1’s comment 6: 

We thank the reviewer for the valuable comment. Changes have been made and the refences have been added as shown below:

Lines 66-70: “In addition to the crucial functions mentioned above, advances in laboratory techniques, the impact of planted forests replacing areas previously used for agriculture and livestock in most cases in Brazil (Rachid et al. 2015; Silveira et al. 2022; Pereira et al. 2021), or afforesting (monoculture or mixed-species) native savannas in current case, on fungal communities are receiving increasing attention in the tropics. 

Silveira, J.G.d.; Oliveira Neto, S.N.d.; Canto, A.C.B.d.; Leite, F.F.G.D.; Cordeiro, F.R.; Assad, L.T.; Silva, G.C.C.; Marques, R.d.O.; Dalarme, M.S.L.; Ferreira, I.G.M.; et al. Land Use, Land Cover Change and Sustainable Intensification of Agriculture and Livestock in the Amazon and the Atlantic Forest in Brazil. Sustainability 2022, 14, 2563. https://doi.org/10.3390/su14052563”

L 54. Reviewer 1’s comment 7 

Why Eucalyptus and/or Acacia? It needs a minimum of two species.

Response to reviewer 1’s comment 7: 

Change has been made: 

Lines 75-76: “Mixed-species plantations of Eucalyptus spp. and Acacia spp. were established

Reviewer 1’s comment 8: 

L 71-74. My perception is that this sentence is confusing and perhaps in the wrong place. Was this territorial extension occupied only by mixed plantations of eucalypt and acacia?

Response to reviewer 1’s comment 8: 

We thank the reviewer for the comment. The territorial extension was occupied by different forest plantations as pines, eucalyptus and acacias. More information about the forest plantation species has been added.

Lines 71-76: “in the Congolese coastal plains has been undertaken, (reaching an overall forested area of around 42.000 ha composed of tropical pines, eucalyptus and acacias in the 2000s), to meet the demand for wood (both in industrial and rural communities), preserve natural forests, mitigate the effects of global warming, and utilize soils unsuitable for agriculture (20)’.

This sentence has been moved at the beginning of the section (lines 79-83).”

Reviewer 1’s comment 9: 

L 79. What type of forest management?

Response to reviewer 1’s comment 9: 

The clarification has been made.

“… in nutrient cycling in the mixed-species forest plantations (line 97).”

Reviewer 1’s comment 10: 

L80-81. Could you specify which forest management and type of stand the authors are comparing? Could you include references to work done elsewhere to support your hypothesis?

Response to reviewer 1’s comment 10: 

We thank the reviewer for the comment and suggestion. Forest management and stand type have been specified. Also references of work conducted elsewhere have been added. 

Lines 99-101: “This study therefore addresses the hypothesis that fungal community composition is linked to acacia and eucalyptus plantations, especially to the mixed acacia and eucalyptus stand as already observed in Brazilian ecosystems (Rachid et al. 2015; Pereira et al. 2021).

Reviewer 1’s comment 11: 

L 82. Don't composition and diversity have the same meaning in the sentence? Wouldn't it be more appropriate to replace it with community structure and diversity?

Response to reviewer 1’s comment 11: 

As suggested, the sentence has been modified:

Lines 102-103:“… on the soil mycobiota community structure and diversity on the sandy and nutrient-poor soils in the Congolese coastal plains.”

Following the suggestion, also the title has been revised: Impact of mixed-species forest plantations on soil mycobiota community structure and diversity in the Congolese coastal plains

Reviewer 1’s comment 12: 

L 89-91. Average air temperature is an important piece of information as well.

Response to reviewer 1’s comment 12: 

We agree with the reviewer. The air temperature and humidity have been added:

Lines 120-121: “season (June to September), and the high mean annual air humidity and temperature (85% and 25 °C, respectively).”

Reviewer 1’s comment 13: 

L 92-93. The first sentence is more in line with the first paragraph.

Response to reviewer 1’s comment 13: 

The following sentence has been moved in the first paragraph as suggested by the reviewer. 

Lines 116-118: “Afforestation using Eucalyptus hybrids on native tropical savanna dominated by the C4 Poaceae Loudetia arundinacea (Hochst.) Steud. occurred in 1984.”

Reviewer 1’s comment 14: 

L94. The description of the experimental device seems confusing to me. Are there three replications with monospecific acacia plantations? And just one stand with eucalypt monoculture? Is it just a stand with mixed planting? Would this be a complete randomized block design?

Response to reviewer 1’s comment 14: 

It is a complete randomized block design. The text was clarified as follow:

Lines 152-155: “Trials (at a density of 800 trees ha-1) were installed on a complete randomized block design in May 2004. The trials consisted of three blocks, each composed of three stands; a monoculture stand of Acacia mangium (100A), a monoculture stand of Eucalyptus urophylla × E. grandis (100E), and a mixed-species stand [50% of Acacia and 50% of Eucalyptus (50A50E)]”.

The reference n. 24 was added. 

Reviewer 1’s comment 15: 

L 98-100. Wasn't there a new mixed planting? It's confusing because it seems that there is no mixed planting for data collection.

Response to reviewer 1’s comment 15: 

The text was rewritten to avoid the confusion:

Lines 125-126: “The first rotation ended in January 2012 (7 years) and the second rotation was established in March 2012 on the same design.”

Reviewer 1’s comment 16: 

L 105. What is the length of the transect? 10 m (similar to the width of the sampling plot)? If that's the case, it seems odd that the sampling points are only 0.7 m apart from each other.

Response to reviewer 1’s comment 16: 

We thank the reviewer for the question. The sentence has been added to clarify the paragraph:

Lines 129-131: “A stand comprised 100 trees (10×10) including 36 trees in an inner part on an area of 1250 m2 and two buffer rows. The spacing between rows was 3.75 m, with 3.33 m between the trees in a row. The spacing between rows was 3.75 m, with 3.33 m between the trees in a row. Three transects in 100A and 100E stands and six in 50A50E stands were set up in the inner part of each stand, starting at the base of a tree and ending in the center of the area delimited by four trees. Three cores were sampled on each transect; each sampling point being separated by 0.7 m from the next along each transect”.

Reviewer 1’s comment 17: 

L110-115. Routine soil fertility practice is to use a 2 mm sieve. Also, it seems confusing to me the way he mixed the simple samples to form the composite samples. In addition, it is unclear how long the sampling and molecular analysis took place, and how the samples were transported from the field to the lab.

Response to reviewer 1’s comment 17: 

We thank the reviewer for the comment. Obviously, the standard sieve is 2mm. However, when it comes to determine particulate organic matter (POM 53-4000µm) the 4mm sieve is used. Soils analyses using POM were conducted to evaluate SOM dynamics (Koutika et al. 2019). As for this study as for other conducted on the same trial design (Koutika et al. 2017; Koutika et al. 2020), composite samples made on the same base have been built. A composite sample has been made from three samples in each stand. Composite soil sampling is the traditional soil sampling method. The objective is to get a sample that represents the average of the area that is being sampled. Three composite samples of pure Acacia (100 A) and Eucalyptus (100 E) and 6 of mixed-species (50 A 50 E) stands were obtained by block, i.e., 12 samples per block and a total of 36 samples for 3 studied blocks. The collected soil samples were transferred to a labelled zipped plastic bag and shipped to ENEA laboratories (Italy) in ice. Once received, samples were stored at -20°C until DNA extraction.

Reviewer 1’s comment 18: 

L117. The elemental analyzer provides the total content of each element. So, it would be appropriate to include the term Total (e.g. Total C, N, and S...).

Response to reviewer 1’s comment 18: 

Line 146: It has been done.

Reviewer 1’s comment 19: 

L129-130. Specify how these samples were distributed among treatments.

Response to reviewer 1’s comment 19: 

Soil was sampled in 9 replicates by stand beneath monoculture of A. mangium (100 104 A) and E. urophylla × E. grandis (100 E) and 18 replicates mixed-species of Acacia and Eucalyptus 105 (50 A 50 E) in 3 blocks (lines 101-105 old version). We better clarified the distribution of samples among treatments as follows:

Lines 134-137: “Nine samples were collected in the 100A and 100E stands and 18 in the 50A50E stands with 9 samples collected near an Acacia, noted 50A50E (near acacia), and 9 others near a Eucalyptus, noted 50A50E (near Euca) in each of the three blocks.

Reviewer 1’s comment 20: 

L 152. Wouldn't a parametric test (ANOVA, Tuckey, etc.) be more appropriate since the data come from a randomized experimental device? Ditto for Spearman's correlation, which could be replaced by Pearson. Has the normality of the data been tested?

Response to reviewer 1’s comment 20: 

Yes, we conducted normality tests on our data using Shapiro-Wilk test (1965). However, field experiments generally present higher variation coefficient. Thus, we chose non-parametric methods where appropriate.

SHAPIRO, S. S., & WILK, M. B. (1965). An analysis of variance test for normality (complete samples). Biometrika, 52(3–4), 591–611. https://doi.org/10.1093/biomet/52.3-4.591

Reviewer 1’s comment 21: 

L 186-189. I don't remember reading that described in the methodology that the samples were collected near eucalypt or acacia trees in mixed plantations.

Response to reviewer 1’s comment 21: 

This remark is correct. We added the following sentence to clarify the soil sampling procedure:

Lines 134-137: “Nine samples were collected in the 100A and 100E stands and 18 in the 50A50E stands with 9 samples collected near an Acacia, noted 50A50E (near acacia), and 9 others near a Eucalyptus, noted 50A50E (near Euca) in each of the three blocks.”

Reviewer 1’s comment 22: 

L 189-192. Both sentences should be in the methodology.

Response to reviewer 1’s comment 22: 

We thank the reviewer for the request. The two sentences have been moved to methodology section. 

Reviewer 1’s comment 23: 

L 192-196. It seems to me that the association of several statistical techniques to present the results is confusing. I miss a greater link between the first sentence (PCA) and the second sentence (Table S2).

Response to reviewer 1’s comment 23: 

We thank the reviewer for this important comment. The link between those 2 sentences was made.

Lines 225-230: “PCA analysis revealed a clear separation of fungal communities in 100E from stands containing Acacia, suggesting the effects of plant species on fungal community structure (Fig 1B). This is confirmed by the statistical analysis reported in S2 Table highlighting a significant difference in fungal beta diversity in mixed-species compared to single stands i.e., (50A50E (near Acacia) vs.100A and 50A50E (near Euca) vs. 100E)”.

Reviewer 1’s comment 24: 

L 217. Report this proportion of unidentified phyla.

Response to reviewer 1’s comment 24: 

The requested proportion has been added: 

Lines 251-252: “On the contrary, the lower relative abundance of the unidentified is apparent in 100E (5%) and higher 7% for both 100A and 50A50E (near Acacia) stands”.

Reviewer 1’s comment 25: 

L231. Authors should avoid single-s

---

## [Editor Report · Decision Letter 1]

29 Aug 2024

Impact of mixed-species forest plantations on soil mycobiota community structure and diversity in the Congolese coastal plains

PONE-D-24-15839R1

Dear Dr. Bevivino

We’re pleased to inform you that your manuscript has been judged scientifically suitable for publication and will be formally accepted for publication once it meets all outstanding technical requirements.

Kind regards,

Tunira Bhadauria, Ph.D.

Academic Editor

PLOS ONE
---

## [Editor Report · Acceptance letter]

30 Sep 2024

PONE-D-24-15839R1 

PLOS ONE

Dear Dr. Bevivino, 

I'm pleased to inform you that your manuscript has been deemed suitable for publication in PLOS ONE. Congratulations! Your manuscript is now being handed over to our production team.

Kind regards, 

on behalf of

Dr. Tunira Bhadauria 

Academic Editor

PLOS ONE